# Fusion with Promiscuous Gα_16_ Subunit Reveals Signaling Bias at Muscarinic Receptors

**DOI:** 10.3390/ijms221810089

**Published:** 2021-09-18

**Authors:** Alena Randáková, Dominik Nelic, Martina Hochmalová, Pavel Zimčík, Mutale Jane Mulenga, John Boulos, Jan Jakubík

**Affiliations:** 1Department of Neurochemistry, Institute of Physiology Czech Academy of Sciences, 14220 Prague, Czech Republic; dominik.nelic@fgu.cas.cz (D.N.); martina.hochmalova@gmail.com (M.H.); senr@seznam.cz (P.Z.); janemulenga14@yahoo.com (M.J.M.); jan.jakubik@fgu.cas.cz (J.J.); 2Department of Physical Sciences, Barry University, Miami Shores, FL 33161, USA; jboulos@barry.edu

**Keywords:** muscarinic receptors, signaling bias, fusion proteins, non-canonical signaling

## Abstract

A complex evaluation of agonist bias at G-protein coupled receptors at the level of G-protein classes and isoforms including non-preferential ones is essential for advanced agonist screening and drug development. Molecular crosstalk in downstream signaling and a lack of sufficiently sensitive and selective methods to study direct coupling with G-protein of interest complicates this analysis. We performed binding and functional analysis of 11 structurally different agonists on prepared fusion proteins of individual subtypes of muscarinic receptors and non-canonical promiscuous α-subunit of G_16_ protein to study agonist bias. We have demonstrated that fusion of muscarinic receptors with Gα_16_ limits access of other competitive Gα subunits to the receptor, and thus enables us to study activation of Gα_16_ mediated pathway more specifically. Our data demonstrated agonist-specific activation of G_16_ pathway among individual subtypes of muscarinic receptors and revealed signaling bias of oxotremorine towards Gα_16_ pathway at the M_2_ receptor and at the same time impaired Gα_16_ signaling of iperoxo at M_5_ receptors. Our data have shown that fusion proteins of muscarinic receptors with α-subunit of G-proteins can serve as a suitable tool for studying agonist bias, especially at non-preferential pathways.

## 1. Introduction

G-protein-coupled receptors (GPCRs) are the largest family of human membrane proteins that transmit signals into a cell through heterotrimeric G-proteins. GPCRs represent the primary target for drug development with potential application in essentially all clinical fields. They mediate a broad range of physiological processes by driving multiple intracellular effectors through various classes of G-proteins. Individual GPCRs preferentially couple to the particular class of G-proteins but they can also successfully activate others [1,2,3]. This coupling promiscuity was observed in both artificial systems with over-expressed GPCRs and native cells [4,5]. Besides G-proteins, GPCR can couple with β-arrestins which desensitize and scaffold G-protein-driven signaling pathways [6]. The multiplicity of signaling leads to the high complexity of the functional response of GPCRs to agonist stimulation.

Structurally different agonists induce specific changes in the GPCRs leading to stabilization of agonist-specific conformations that can lead to non-uniform agonist-specific modulation of signaling pathways. This preferential orientation of the signaling of a GPCR towards a subset of its signal transducers is termed signaling bias [7]. An agonist biased to a particular G-protein pathway may promote therapeutically desired signaling while simultaneously avoiding side effects mediated by activation of others, especially in conditions with well-understood pathophysiology [8,9,10]. For example, melanocortin receptor 4 (MC4R) agonist melanotan II produces its anorectic effects through coupling to G_q/11_ and its adverse cardiovascular effects through G_s_ coupling, suggesting potential therapeutic benefit in obesity for G_q/11_-biased ligands [11].

The accurate evaluation of agonist bias regarding individual G-protein pathways is crucial for preclinical drug development. However, it is a difficult task given the high complexity of GPCRs signaling. The molecular crosstalk that can occur among downstream effector molecules may bring in further complexity [12]. One of the most challenging tasks is to develop a suitable technique for the analysis of the signaling pathway of interest with high sensitivity and sufficient selectivity that is free from the interference of other signaling pathways. Measurement of second messengers struggles with molecular crosstalk of signaling pathways. The analysis of coupling of GPCRs with individual G-proteins is difficult due to the presence of others that interact concurrently with a given signaling pathway, especially in studies of non-preferential signaling pathways [13] or in studies of signaling pathways mediated by individual isoforms of given G-protein. 

Muscarinic signaling is implicated in numerous pathologic events, such as the promotion of carcinoma cell growth, early pathogenesis of neurodegenerative diseases in the central nervous system of Alzheimer’s and Parkinson’s, schizophrenia, drug addiction, pain, and also in some internal diseases, e.g., asthma or overactive bladder [14,15]. As of now, no affinity-based selective agonists of individual muscarinic receptors have been discovered, due to the high homology of the orthosteric binding site among individual muscarinic subtypes [16,17,18]. 

Selective targeting on the G_i/o_ versus G_q/11_ mediated pathway by biased agonist could be a way to achieve selectivity to even or odd muscarinic subtypes [19]. Moreover, agonists biased to individual isoforms of G-proteins could lead to tissue-specific activation of mAChRs, due to the predominant expression of some G-proteins in specific tissues (e.g., G_o_ in the central nervous system or G_16_ in hematopoietic cells) [10,20]. We have focused on variations in the G_16_ signaling pathway that was not studied so far, is rare and may lead to very specific effects (e.g., tissue-specific activation). We have analyzed variation in the G_16_ signaling profile among individual subtypes of muscarinic receptors.

To reveal and properly quantify putative agonist bias to certain G-proteins and their isoforms, especially non-preferential ones, among individual subtypes of muscarinic receptors, a system that is sufficiently sensitive and specific is required. Furthermore, 1:1 Gα-receptor stoichiometry would simplify the analysis and interpretation of found agonist bias. We assume that fusion proteins of muscarinic receptors with Gα subunit of interest could serve as a convenient tool to screen agonist bias towards the particular Gα among individual subtypes of muscarinic receptors. Importantly, we expect that tight fusion of a receptor with a particular Gα prevents the coupling of other competing Gα to the receptor. If so, the signaling of a pathway of interest can be selectively analyzed. Fusion proteins of GPCR and α-subunit of G-protein were used to study individual G-protein mediated pathways in several studies [21,22,23,24,25,26]. We validate our assumptions on an example of fusion proteins of individual muscarinic receptors and non-canonical promiscuous Gα_16_ subunit.

Muscarinic signaling via non-canonical G_16_ G-protein may play a relevant physiological role. At the protein level, G_16_ expression is only detected in highly specific cell types (hematopoietic and epithelial cells) characterized by a high rate of cell turnover [27]. G_16_ mediated signaling may play role in immune response [28] and tumor cell growth [29].

Promiscuous Gα_16_ efficiently couples to any subtype of muscarinic receptor. That leads to the phospholipase C-activation, resulting in the formation of inositol phosphates. We performed a binding and functional analysis of these constructs using eleven structurally different muscarinic agonists. We demonstrated agonist-specific activation of non-canonical G_16_ pathway varying among individual subtypes. Additionally, we compared the signaling of agonists oxotremorine and iperoxo at Gα_16__fused, Gα_16_ co-transfected, and wild types of M_2_ and M_5_ muscarinic receptors and revealed signaling bias of oxotremorine towards Gα_16_ pathway at the M_2_ receptor and at the same time impaired Gα_16_ signaling of iperoxo at M_5_ receptors.

## 2. Results

### 2.1. Fusion Proteins

#### 2.1.1. Description of Fusion Proteins

Fusion proteins (denoted M_1__Gα_16_ through M_5__Gα_16_) were constructed from individual subtypes of muscarinic receptors M_1_–M_5_ and α-subunit of G_16_ G-protein. The α-subunit was tightly connected to the C-terminus of the respective receptors as described in the Methods. Palmitoylation sites at helix 8 of receptors, as well as at N-terminus of Gα_16_, were preserved to ensure their anchoring to the membrane. Complete sequences of fusion proteins are shown in Appendix A. 

#### 2.1.2. Homology Models of Fusion Proteins

To test whether fusion proteins respect the natural arrangement of the receptor and G-protein α-subunit that allows their successful coupling with no serious occurring disturbance to the structure arrangement, we have built homology models of M_1__Gα_16_ and M_2__Gα_16_ fusion proteins. Homology modeling resulted in a good model free of unusual structural features. Overlays of fusion proteins with cryo-EM of M_1_ + Gα_11_ (6OIJ) and M_2_ + Gα_o_ (6OIK) receptor-G-protein complexes [30] are shown in Figure 1. The stability of structure was verified by running molecular dynamics (MD) of fusion proteins in complex with G-protein βγ-dimer in membrane/water system. Analysis of MD trajectories by Simulation Quality Analysis tools of Maestro confirmed the stability of the structures (Appendix A). No structural rearrangements occurred during 120 ns of MD. Insertion of the C-terminus of Gα_16_ to G-protein binding site at the receptor located between transmembrane helix 3 and 6 corresponds to the insertion of Gα_11_ at M_1_ and Gα_o_ at M_2_. At the M_1_ receptor, the position of the C-terminus of Gα_11_ and Gα_16_ are practically identical. However, at M_2_, Gα_16_ is inserted under a sharper angle than Gα_o_. Insertion of the C-terminus of Gα_16_ instead of Gα_11_ or Gα_o_ to the G-protein binding site did not induce any major change in the receptor conformation.

#### 2.1.3. Affinity of [^3^H]NMS for Gα_16_ Fused Receptors

To confirm that fusion with Gα_16_ indeed did not influence receptor conformation as indicated by molecular modeling, we measured the binding of radiolabeled [^3^H]NMS to all Gα_16__fused receptors. The affinity of [^3^H]NMS to fusion proteins was determined in saturation binding experiments and calculated according to Equation (1). The fusion of muscarinic receptors with Gα_16_ subunit did not affect the binding affinity of [^3^H]NMS at any fusion protein. The determined affinity of [^3^H]NMS to fused and wt receptors, as well as their expression level in CHO cells, is summarized in (Appendix A). 

### 2.2. Lack of Coupling of Gα_16__Fused Receptors with Endogenous G-Proteins

Muscarinic receptors are able to activate multiple G-proteins. Preferentially, muscarinic receptors M_1_, M_3_, and M_5_ couple with Gα_q/11_ and M_2_ and M_4_ receptors with Gα_i/o_ G-proteins. All muscarinic subtypes efficiently activate non-canonical promiscuous G-protein (G_16_) followed by activation of the appropriate signaling pathway (phospholipase C-activation and generation of IPX). Based on molecular models, we expected that fusion of muscarinic receptors with Gα_16_ subunit would sterically prevent coupling of endogenously expressed G-proteins. To this end, we analyzed changes in cAMP level, mediated by endogenous G_i/o_ and G_s_ proteins, after activation of wt and Gα_16__fused M_2_ and M_4_ receptors by agonist carbachol. Basal level of cAMP was determined in presence of 10 µM adenylate cyclase activator forskolin. Values of basal level determined as % of incorporated radioactivity varied in the range of 2.5–3% and are the same in cells expressing wt and fused receptors. Level of cAMP was calculated as fold over basal level. We demonstrate that tight fusion with Gα_16_ prevented the coupling of preferential Gα_i/o_ and non-preferential G_s_ to M_2_ and M_4_ receptors. While carbachol stimulated accumulation of IP_X_ at all Gα_16__fused receptors (Appendix A), at M_2__Gα_16_ and M_4__Gα_16_, did not change the level of cAMP, whereas at wt M_2_ and M_4_, carbachol inhibited cAMP synthesis via preferential G_i/o_ G-proteins at submicromolar concentrations and stimulated it via non-preferential G_s_ G-proteins at micromolar concentrations (Figure 2). Thus, the fusion proteins pass the signal solely through the fused Gα subunit. 

### 2.3. Binding and Functional Analysis of Gα_16_ Fused Receptors

Eleven structurally different agonists, varying in the binding mode to muscarinic receptors, potency, and efficacy to activate muscarinic receptors (arecoline, carbachol furmethide, iperoxo, McN-A343, N-desmethylclozapine, oxotremorine, pilocarpine, xanomeline, JR-6, and JR-7), were used for pharmacological evaluation of the fusion proteins. Structures of tested agonists are shown in Appendix A. 

#### 2.3.1. Binding Affinity of Tested Agonists to Gα_16_ Fused Muscarinic Receptors

The affinity of tested agonists to fused proteins was assayed in competition experiments with 1nM [^3^H]NMS, calculated according to Equation (4), and is summarized in Table 1. All agonists completely inhibited [^3^H]NMS binding to fused proteins. All tested agonists displayed only low-affinity binding, except for iperoxo at M_1__Gα_16_ and JR6 at M_4__Gα_16_. Affinities of low-affinity binding of tested agonists (carbachol, oxotremorine, pilocarpine, JR6, and JR7) to wt and Gα_16__fused muscarinic receptors were compared. Data are summarized in Appendix A. The affinity of carbachol was slightly lower at all Gα_16__fused receptors than at corresponding wt. The decrease in affinity was observed also for pilocarpine, JR7, and oxotremorine (except oxotremorine at M_1_ and JR7 at M_2_). On the other hand, JR6 had a higher affinity at all Gα_16__fused receptors, especially at M_2__Gα_16_ affinity of JR6 was 23-times higher than at wt M_2_.

#### 2.3.2. Functional Response of Gα_16__Fused Muscarinic Receptors to Agonists

The fusion with non-canonical promiscuous G-protein Gα_16_ couples all subtypes of muscarinic receptors to phospholipase C-activation and generation of IP_X_. The level of IP_X_ was measured by radio-chromatographic separation. Basal level (in absence of agonist) varied in range of 2–3% of incorporated radioactivity and was the same in cells expressing wt and individual fused receptors. Level of IP_X_ in presence of individual concentrations of tested agonists was calculated as fold over basal level. Parameters of accumulation of IP_X_ as a functional response of fused proteins to stimulation by a tested agonist, EC_50_ and E′_MAX_, are summarized in (Appendix A). To calculate the coefficient of operational efficacy τ of functional response of individual Gα_16__fused receptors to tested agonists, the system E_MAX_ was determined from functional responses to the agonists carbachol, oxotremorine, and pilocarpine according to the procedure described recently [31]. Then, the τ value was used for the calculation of the equilibrium dissociation constant K_A_. The values of τ and K_A_ calculated according to Equation (6) are summarized in Table 2. 

G_i/o_-biased muscarinic partial agonists JR6 and JR7 did not stimulate the accumulation of IP_X_ at any fused protein. Although fusion with Gα_16_ led to an increase in the affinity of JR6 to all fusion proteins, JR6 and JR7 induced conformation incompatible with activation of the Gα_16_ signaling pathway. Except for JR6 and JR7, all tested agonists stimulated accumulation of IPx at all Gα_16__fused receptors. 

Quantification of Agonist Bias towards Individual Gα_16_ Fused Receptors.

To compare agonist specific activation of G_16_ mediated pathway among individual muscarinic subtypes and to quantify agonist bias towards individual Gα_16__fused receptors, intrinsic activities relative to carbachol (RA_i_) were calculated according to Equation (8) from the E´_MAX_ and EC_50_ values of accumulation of inositol phosphates (Appendix A). Values of RA_i_ are summarized in Table 2 and plotted in Figure 3. Interestingly, M_2_ super-agonist iperoxo [32,33] displayed a strong bias to M_3__Gα_16_ over the rest of the subtypes. Iperoxo RA_i_ values for other Gα_16__fused receptors were two (M_2_) to 20-fold (M_4_) lower. On the other hand, N-desmethylclozapine, considered as M_1_ preferring agonist [34] displayed bias to M_1__Gα_16_ and M_3__Gα_16_ over the rest of the subtypes. The most pronounced bias was found in the case of McN-A-343. The RA_i_ for M_2__Gα_16_ was more than 30-times higher than RA_i_ for M_3__Gα_16_. On the other hand, signaling profiles to individual Gα_16__fused receptors of ligands like xanomeline, oxotremorine, pilocarpine were almost balanced. The majority of agonists (arecoline, furmethide, McN-A-343, pilocarpine, xanomeline, and oxotremorine) displayed bias to M_2__Gα_16_. RA_i_ of arecoline decreases in order M_2_ > M_4_ ≈ M_3_ > M_5_ > M_1_, RA_i_ of furmethide in order M_2_ > M_5_ ≈ M_4_ > M_1_ > M_3_, McN-A-343 M_2_ > M_4_ > M_5_ > M_1_ > M_3_, pilocarpine M_2_ > M_5_ > M_3_ ≈ M_1_ > M_4_, xanomeline M_2_ > M_5_ > M_4_ > M_1_ > M_3_, and oxotremorine M_2_ > M_5_ > M_1_ ≈ M_3_ > M_4_. The same results were obtained using quantification of signaling bias by calculation of bias factor10^∆∆log(τ/KA)^ introduced by Kenakin et al., 2012 [35] (Appendix A). The variability in bias among agonists eliminates the possibility that protein fusion introduced a bias towards some of the receptors.

Functional Response of Gα_16__Fused, Gα_16_ co-Transfected Receptors, and wt Receptors to Selected Agonists.

We analyzed activation of IP_X_ pathway by agonist carbachol, oxotremorine, and iperoxo at Gα_16__fused receptors, receptors co-transfected with Gα_16_-subunit and wt M_2_ and M_5_ receptor (Figure 4, Table 3). 

Agonist induced coupling with Gα_16_**:** Data show better coupling of Gα_16_ fused M_2_ receptor in comparison to the co-transfected system for carbachol and oxotremorine. At IP_X_ pathway, the value of equilibrium dissociation constant expressed as the negative logarithm (pK_A_) for reference agonist carbachol as well as tested agonist oxotremorine, was higher at Gα_16__fused receptors than at M_2_ receptors co-transfected with Gα_16_ (Table 3), indicating a better coupling in the case of the fusion protein. The better coupling to G_16_ at fused receptors M_1__Gα_16_ through M_5__Gα_16_ than at co-transfected variants is obvious also from comparison of (Table 2 with our previous data Randakova et al. [28]). The pK_A_ value for reference agonist carbachol and oxotremorine, as well as pilocarpine or xanomeline, was higher at Gα_16__fused receptors than at corresponding wt receptors co-transfected with Gα_16._ The increase in pK_A_ ranged from 3-fold for xanomeline at M_1_ to 63-fold for oxotremorine at M_3_.

In contrast, pK_A_ of iperoxo at the fused M_5__Gα_16_ was lower than at co-transfected M_5_+Gα_16_, indicating worse coupling of the fusion protein (Table 3). The high variability in the observed shift in pK_A_ excludes a possibility of the systemic artifact caused by protein fusion.

Comparison of operational efficacies of selected agonists: In comparison to the co-transfected system M_2_+Gα_16_, the increase in operational efficacy τ of both reference agonist carbachol as well as tested agonist oxotremorine to stimulate the non-canonical accumulation of IP_X_ at M_2__G_α16_ (Table 3) indicates the higher sensitivity of measurement of functional response at Gα_fused receptors. Oxotremorine had higher operational efficacy than carbachol at M_2__Gα_16_ and M_5__Gα_16_ (Table 2). At the rest of the Gα_16__fused receptors, the operational efficacies of oxotremorine and carbachol were the same. In contrast, oxotremorine stimulated accumulation of IP_X_ at M_2_ + Gα_16_ and M_2_ wt with efficacy comparable (Table 3) or lower [19] to carbachol. Operational efficacies τ of functional responses of carbachol and oxotremorine at M_2__Gα_16_ and M_2_ + Gα_16_ (Figure 4) are summarized in Table 3. In other words, at M_2__G_α16_ fusion protein (where M_2_ receptor signals only via G_α16_) oxotremorine had higher efficacy than in co-transfected system (where binding of other Gα subunits to M_2_ may take place) which indicates bias of oxotremorine towards Gα_16_ mediated pathway at M_2_ receptor.

Interestingly, agonist iperoxo had higher operational efficacy τ than carbachol at all Gα_16__fused receptors, except M_5__Gα_16_ (Table 2). At fused M_5__Gα_16_, τ value of iperoxo was almost 30% lower than τ values of carbachol. At the rest of Gα_16__fused receptors, τ values of iperoxo were greater than τ for carbachol, least by 40 % (M_4_) and most nearly 7-fold (at M_2_). In contrast to M_5__Gα_16_, iperoxo stimulated accumulation of IP_X_ at M_5_-wt with higher operational efficacy than carbachol. At co-transfected system M_5_ + Gα_16_, iperoxo and carbachol stimulated IP_X_ accumulation with comparable efficacy (Figure 4, Table 3) which indicates impairment of Gα_16_ signaling of super-agonist iperoxo at fused M_5_ receptor. 

## 3. Discussion

In this study, we show that fusion proteins of receptor and α-subunit of G-protein are a suitable tool for studying agonist bias. We demonstrate it on the example of muscarinic receptors fused with Gα_16_ subunit and 11 muscarinic agonists whose signaling profile (bias) varies among receptor subtypes. 

Analysis of signaling bias of muscarinic receptors, concerning G-protein mediated signaling, has several pitfalls. Coupling promiscuity of muscarinic receptors leads to molecular crosstalk in downstream signaling. For example, calcium ions released upon activation of G_q/11_ IP_X_ pathway modulate some adenylate cyclases and thus cAMP signaling. In turn, βγ-dimers released from G_i/o_ G-proteins modulate some calcium channels and thus calcium signaling [36,37]. Moreover, signals of non-preferential pathways are usually weak, thus, highly sensitive methods are needed. The main obstacle, in the study of the non-preferential G-protein pathways, is the competition of different (mainly preferential) Gα-subunits for the binding site at a given receptor. Activation of a non-preferential G-protein pathway may play important roles in processes characterized by fluctuation in an expression of individual G-proteins or GPCRs, e.g., immune cell maturation [28], progression of cancer [38], or Parkinson´s disease [39].

Several tools including G-protein-specific pharmacological inhibitors or toxins [40], C-terminus mimicking peptides [41], small interfering RNA [42,43], using artificial systems with limited endogenous G-proteins [44,45,46] or reconstitution of purified receptors and G-proteins in the artificial membrane [47,48] limit the signal mediated by certain G-proteins. Techniques like the immunoprecipitation with specific Gα antibodies [2,49], resonance energy transfer techniques, where bioluminescent (BRET) or fluorescent (FRET) donors and acceptors are fused on the C-terminus of the GPCR and in one of the subunits of the G-protein [50,51] were used to study specific GPCR-G-protein interactions. Although these methods diminish or eliminate signaling crosstalk, they are not aimed at high sensitivity.

Receptor_Gα fusion proteins are well described to study the activation of individual G-protein mediated signaling pathways at many GPCR [21,22,23,24,25,26]. We demonstrate their use to study agonist bias at non-canonical G_16_ pathway among individual subtypes of muscarinic receptors. Gα_16(15)_ is expressed only in highly specific cell types such as hematopoietic and epithelial cells [27], which are characterized by a high rate of cell turnover. Muscarinic receptors expressed in these cells appear to be involved in the regulation of diverse cellular activities including immune response [28], cell proliferation, or cell differentiation [52,53].

The engineering of receptor-transducer fusion proteins seems to be an effective strategy to target cellular effectors more efficiently and specifically [21]. Fusion proteins enable the study of signaling mediated by G-proteins up to the level of individual G-proteins isoforms. Moreover, receptor-G-protein fusion forces a 1:1 stoichiometry and ensures efficient coupling of the given receptor to an attached Gα subunit. Receptor-G-protein stoichiometry is a relevant aspect of signaling bias and should be taken into account in the screening of biased agonists [54].

We have created fusion proteins of individual muscarinic receptors (M_1_–M_5_) and non-canonical promiscuous Gα_16_ subunit and performed detailed binding and functional analysis of these constructs using 11 structurally different muscarinic agonists to evaluate the suitability of such fusion proteins to study agonist bias. Structurally different agonists vary in interactions in the orthosteric binding site of the muscarinic receptor [55]. The portfolio of used agonists included reference balanced full agonist carbachol, classic muscarinic agonists arecoline, furmethide, pilocarpine, oxotremorine, super-agonist iperoxo [32,33], bitopic agonists xanomeline [56], and McN-A343 [57], and Gi/o-biased agonists JR6 and JR7 [19] (Appendix A). 

The use of Gα_fused receptors for analysis of signaling bias is conditioned by the full preservation of binding and functional properties of both receptor and Gα subunit. In the preparation of the construct, palmitoylation sites, at the C-terminus of the receptors [58], and the N-terminus of the Gα subunit [59], that mediate interaction with the membrane, were maintained. That is essential for keeping the native conformation of a receptor as well as G-protein. Comparison of homology models of prepared constructs M_1__Gα_16_ and M_2__Gα_16_ with cryo-EM structures of receptor-G-protein complexes M_1_ + Gα_11_ and M_2_ + Gα_o_ [30] confirmed the natural arrangement of the receptor and Gα in fusion proteins (Figure 1). At the M_1_ receptor, insertion of the C-terminus of related Gα_16_ and Gα_11_ subunits into the G-protein binding site of the receptor is practically identical. On the other hand, at the M_2_ receptor, evolutionarily more distant Gα_16_ and Gα_o_ differ in the angle at which they are inserted into the G-protein binding site. Gα-specific insertion of C-terminus into the intracellular cavity of cognate GPCR was observed in 3D structures of GPCR-G-protein complexes [30,60,61,62,63] and demonstrated using molecular dynamics (MD) as well [64]. Furthermore, the fusion of muscarinic receptors with Gα_16_ subunit did not affect the binding affinity of the labeled antagonist [^3^H]N-methylscopolamine at any fusion protein (Appendix A), indicating that fusion did not markedly influence receptor conformation. 

The signaling of interest can be selectively analyzed when the binding of other competing G-proteins to the receptor is excluded. We hypothesized that tight fusion of the receptor with a particular Gα subunit prevents the binding of other G-proteins. The M_2_ and M_4_ receptors preferentially inhibit cAMP synthesis via Gα_i/o_ G-proteins and can also couple with non-preferential Gα_s_ to activate cAMP synthesis [65,66]. In contrast to the wt M_2_ and M_4_ receptors (Figure 2), carbachol did not induce changes in cAMP level at fused M_2__Gα_16_ and M_4__Gα_16_ receptors, indicating no coupling to endogenous G_i/o_ or G_s_ G-proteins. It suggests that, unlike some fusion constructs [67], our directly Gα_16__fused constructs indeed prevent the access of competitive Gα subunits to the receptor. 

The binding analysis has shown that in contrast to wild-type (wt) receptors, at Gα_16__fused constructs, almost all tested agonists displayed only low-affinity binding. G-protein binding to a receptor might, in turn, allosterically influence ligand binding [68,69]. The absence of high-affinity binding of most agonists to Gα_16__fused receptors may be either due to lack of pre-coupling of Gα_16_ to the receptor or receptor is pre-coupled Gα_16_ that binds GDP [43]. Since the decrease in the value of equilibrium dissociation constant (K_A_) of agonists at Gα_16__fused receptors in comparison to wt receptor co-expressed with Gα_16_ (Table 3, Table 2 versus our previous data [28]) indicates pre-coupling, the absence of high-affinity binding indicates pre-coupling Gα_16_ that binds GDP [43].

G_16_ G-protein is efficiently capable to couple all muscarinic subtypes (M_1_–M_5_) via phospholipase C activation (IP_X_ accumulation). Thus, it may be possible to analyze the activation of all muscarinic subtypes using one assay (measurement of the accumulation of inositol phosphates, IP_X_) and demonstrate agonist-specific activation of this pathway. Signaling bias among individual Gα_16__fused receptors was calculated from relative intrinsic activities RA_i_ to reference agonist carbachol [70] (Table 2). RAi values can be easily calculated for several pathways and many ligands and quickly compared. In principle, for a single signaling pathway and two or more receptors, a ligand that has greater RAi at one receptor than at other(s) is biased to a given pathway at that receptor. Additionally, we analyzed our data also by conventionally used bias factor [35]. Data are summarized in (Appendix A) and plotted (Appendix A). Quantification of agonist bias obtained by both ways was the same, showing that a quick comparison of RAi factors is sufficient and that analysis was conducted correctly. Presented data demonstrate differences in the pattern of the Gα_16_ pathway activation at five subtypes of Gα_16__fused muscarinic receptors after stimulation by structurally different agonists. It points to variations in the compatibility of agonist-specific conformations with Gα_16_ coupling and activation. While some agonists have quite balanced Gα_16_ pathway activation patterns—such as pilocarpine, oxotremorine, or xanomeline—profound bias towards individual Gα_16__fused muscarinic receptors was observed for agonists McN-A-343 (towards M_2__Gα_16_) and iperoxo (towards M_3__Gα_16_). McN-A343 is a bitopic agonist, capable of stimulating the G_q_ pathway while incapable of stimulating G_s_ at M_1_ expressed in CHO cells [64]. We have shown that McN-A343 successfully activates G_16_ pathway at all muscarinic subtypes with a bias towards M_2_. Interestingly, M_2_ super-agonist iperoxo displayed bias towards M_3__Gα_16_ over other Gα_16__fused receptors. It was demonstrated that iperoxo-based dualsteric compounds exert bias G_i/o_ over G_s_ pathway at M_2_ [71] but exert bias to G_q/11_ over G_i/o_ signaling at the M_1_ receptor [72]. On the other hand, M_1_-preferring agonist N-desmethylclozapine displayed bias to M_1__Gα_16_ and M_3__Gα_16_ over the rest of the subtypes. It points to huge variability in signaling depending on the combination of a ligand–receptor-pathway system, promising a chance to find agonists with a bias to the desired pathway at the desired receptor subtype.

Comparison of parameters of functional response of selected agonists at Gα_16__fused and Gα_16_ cotransfected wt receptors suggest better coupling of fused Gα subunit. The better coupling of Gα_16_ in fusion proteins was demonstrated by a decrease in the value of equilibrium dissociation constant (K_A_) of agonists at Gα_16__fused receptors (Table 2 vs. our previous data Randakova et al. [19], Table 3), except iperoxo at M_5__Gα_16_ (Table 3, discussed below). The elimination of interaction with other competitive Gα subunits as well as fusion alone could lead to better coupling of fused Gα subunits. The operational equilibrium dissociation constant K_A_ quantifies an affinity of agonist to the conformation that initiates a given signaling pathway. Thus, it can be considered as one of the coupling parameters.

Furthermore, the operational efficacy τ to stimulate the non-canonical accumulation of IP_X_ induced both by reference agonist carbachol and tested agonist oxotremorine is higher at fusion protein M_2__Gα_16_ than at co-transfected system M_2_ + G_α16_ (Table 3). The better coupling (both the decrease in K_A_ and increase in τ) indicates that the fusion protein strategy is highly sensitive and thus suitable for detection and analysis of low-efficacy pathways.

Despite the high sensitivity, we did not detect accumulation of IPx induced by G_i/o_ biased agonist JR6 and JR7 (Table 2) at any fused protein. These data further support the true G_i/o_ bias of these novel agonists and also support the suitability of these fusion systems in the analysis of signaling bias.

Our data demonstrate that oxotremorine stimulates accumulation of IPx at M_2__Gα_16_ more efficiently in comparison with co-transfected system M_2_+Gα_16_, where the competition of endogenous G_i/o_ and G_q/11_ occurs and more efficiently than at wt M_2_ via endogenous G_q/11_ (Table 3). In our previous study of Randáková et al. [19], oxotremorine displayed lower RA_i_ to stimulate the accumulation of IP_X_ in the co-expressed system M_2_ + Gα_16_ than in the presented study. This discrepancy can be explained by different levels of expression of Gα_16_ in co-expressed systems and points to the advantage of using fusion proteins with 1:1 stoichiometry for easier spotting of agonist bias. In comparison with our previous data [19], oxotremorine exerts bias towards IPx accumulation (via M_2__Gα_16_) over cAMP inhibition via G_i/o_ at wt M_2_. Signaling bias of agonist oxotremorine to G_16_ over G_i/o_ and G_q/11_ pathway at M_2_ receptor would be hard to reveal and quantify without fusion proteins due to signaling crosstalk or could be overlooked due to competition with other G-proteins. We show that using fusion proteins for this analysis can be very practical.

Furthermore, we demonstrate impairment of Gα_16_ signaling of super-agonist iperoxo at the M_5_ receptor. Besides worse coupling (lower pK_A_) of iperoxo to fused M_5__Gα_16_ (Table 3), unlike other Gα_16__fused receptors, super-agonist iperoxo stimulated accumulation of IP_X_ at M_5__Gα_16_ with lower operational efficacy than reference agonist carbachol. In contrast at wt M_5_ receptors expressed in CHO cells, iperoxo stimulated accumulation of IP_X_ through cognate Gα_q/11_ with higher operational efficacy than carbachol. In CHO cells expressing wt M_5_ co-transfected with Gα_16_, operational efficacy for carbachol and iperoxo was the same (Figure 4, Table 3), which could be explained by competition of Gα_16_ with endogenous preferential G_q/11_. Combined data thus indicate incompatibility of active M_5_ receptor conformation specific to iperoxo with Gα_16_ coupling and activation.

## 4. Materials and Methods

### 4.1. Construct Preparation

Constructs containing sequences of human variants of muscarinic acetylcholine receptors M_1_–M_5_ fused with the human variant of Gα_15_ subunit (also known as Gα_16_ [73]) were prepared, and new stable cell lines of Chinese hamster ovary (CHO) expressing these fusion proteins were generated. Plasmids pcDNA3.1 coding human receptors M_1_–M_5_ and Gα_16_ subunit were obtained from Missouri S&T cDNA Resource Center (Rolla, MO, USA). Plasmid pCMV6-A-Hygro containing hygromycin as a mammalian selection marker was purchased from Origene (Rockville, MD, USA). The coding sequence for Gα_16_ subunit and subsequently sequences for M_1_–M_5_ receptors and were subcloned into the pCMV6-A-Hygro vector using restriction endonucleases. To this end, restriction site AflII at the N-terminus of the Gα_16_ subunit and AgeI at the C-terminus of receptor sequences were created. Both parts were connected via short GATRARS linker, corresponding to the C-terminal amino acids in the M_2_ sequence and N-terminal amino acid of the Gα_16_ subunit. Cysteines needed for palmitoylation of receptors (C^435^ at M_1_, C^457^ at M_2_, C^561^ at M_3_, C^470^ at M_4_, and C^512^ at M_5_) were preserved. Sequences of all fusion proteins are in the Appendix A.

### 4.2. Homology Modeling

Homology models of fusion proteins were constructed as hybrid models using YASARA software, Biosciences (Vienna, Austria) [74]. For M_1__Gα_16_ fusion protein structures PDB ID: 6WJC, 5CXV, 3SN6, 6OIJ, and 6PT0 were selected by the program as templates. For M_2__Gα_16_ fusion protein structures PDB ID: 5ZK3, 6OIK, 3SN6, 6OIJ, and 6PT0 were selected by the program as templates. Modeling parameters were set as follows:

Modeling speed: Slow.

Number of PSI-BLAST iterations in template search: 4.

Maximum allowed (PSI-)BLAST E-value to consider template: 0.5.

Maximum number of templates to be used: 5.

Maximum number of templates with the same sequence: 1.

Maximum oligomerization state: 4 (tetrameric).

Maximum number of alignment variations per template: 5.

Maximum number of conformations tried per loop: 50.

Maximum number of residues added to the termini: 10.

### 4.3. Molecular Dynamics

The homology model of fusion proteins and structure of M_1_ receptor in complex with G_11_ G-protein (6OIJ) were aligned on the receptor part using MUSTANG [75]. The βγ-dimer from the 6OIJ structure was added to the homology model. To evaluate the stability of homology models, conventional molecular dynamics (MD) was simulated using Desmond/GPU ver. 6.1, D. E. Shaw Research (New York, NY, USA). The simulated system consisted of a receptor–G-protein complex in 1-palmitoyl-2-oleoyl-sn-glycero-3-phosphocholine membrane set to receptor helices in water and 0.15 M NaCl. The system was first relaxed by the standard Desmond protocol for membrane proteins. Then 120 ns of NPγT (Nosé–Hoover chain thermostat at 300 K, Martyna–Tobias–Klein barostat at 1.01325 bar, isotropic coupling, Coulombic cut-off at 0.9 nm) molecular dynamics without restrains was simulated. The quality of molecular dynamics simulation was assessed by Simulation Quality Analysis tools of Maestro.

### 4.4. Cell Culture and Membrane Preparation

CHO-K_1_ cells, ATCC (Manassas, VA, USA) were transfected with the desired plasmids using Lipofectamine 3000, Invitrogen (Carlsbad, CA, USA). Subconfluent cells were washed with phosphate-buffered saline and then Opti-MEM, Life Technologies (Carlsbad, CA, USA) containing Lipofectamine at a final concentration of 5 μL/mL and plasmid DNA at a final concentration of 1 μg/mL was applied. After 48 h cells were diluted 1000-times by subculturing and hygromycin-B, Toku-E (Bellingham, WA, USA) was added at a final concentration of 200 µg/mL for selection of transfected clones. Selected clones of each construct were used up to passage 10. The expression level of fused muscarinic receptors was confirmed in radioligand binding experiments using ^3^H-N-methylscopolamine ([^3^H]NMS),ARC (ST.Louis, MO, USA). Additionally, CHO-K1 cells were also transiently co-transfected with plasmids coding muscarinic receptors and plasmid coding Gα_16_ subunit. For transient transfection, linear polyethyleneimine PEI 25K, Polysciences, (Hirschberg, Germany) was used. Subconfluent cells were incubated 24 h in the growth medium containing PEI at a final concentration of 2.4 μg/mL and plasmid DNA at a final concentration of 0.8 μg/mL. After 24 h, fresh medium was added, and cells were harvested 48 h after transfection.

CHO cells expressing individual Gα_16__fused muscarinic receptors were grown to confluence in 75 cm^2^ flasks in Dulbecco’s modified EaglE′s medium supplemented with 10% fetal bovine serum, Life Technologies (Carlsbad, CA, USA). One million cells were subcultured in 100 mm Petri dishes. Cells were washed with phosphate-buffered saline and harvested by mild trypsinization for functional experiments or manually for binding experiments on day five after subculture. After harvesting cells were centrifuged for 3 min at 250× *g*.

Membranes from CHO cells were prepared for binding experiments. The pellets of harvested cells were suspended in the ice-cold homogenization medium (100 mM NaCl, 20 mM Na-HEPES, 10 mM EDTA, pH = 7.4) and homogenized on ice by two 30 sec strokes using a Polytron homogenizer Ultra-Turrax; Janke & Kunkel GmbH & Co. KG, IKA-Labortechnik, (Staufen, Germany) with a 30-sec pause between strokes. Cell homogenates were centrifuged for 5 min at 1000× *g*. The supernatant was collected and centrifuged for 30 min at 30,000× *g*. Pellets were suspended in the washing medium (100 mM NaCl, 10 mM MgCl2, 20 mM Na-HEPES, pH = 7.4), left for 30 min at 4 °C, and then centrifuged again for 30 min at 30,000× *g*. The resulting membrane pellets were kept at −80 °C until assayed.

### 4.5. Radioligand Binding Experiments

All radioligand binding experiments were optimized and carried out as described by El-Fakahany and Jakubik [76]. Briefly, membranes (approximately 10 μg of membrane proteins per sample) were incubated in 96-well plates for 3 h at 25 °C in the incubation medium (100 mM NaCl, 20 mM Na-HEPES,10 mM MgCl_2_, pH = 7.4). In the case of the M_5_ receptor, which has very slow kinetics of binding, the incubation time was extended to 5 h. Incubation volume for competition and saturation experiments with [^3^H]NMS was 400 μL or 800 μL, respectively.

In saturation experiments of binding of [^3^H]NMS six concentrations of the radioligand (ranging from 63 to 2000 pM) were used. Agonist binding was determined in competition experiments with 1 nM [^3^H]NMS. Nonspecific binding was determined in the presence of 10 μM unlabeled atropine. Incubation was terminated by filtration through Whatman GF/C glass fiber filters, Whatman (Maidstone, GB using a Brandel harvester, Brandel (Gaithersburg, MD, USA). Filters were dried in a microwave oven (3 min, 800 W), and then solid scintillator Meltilex A was melted on filters (105 °C, 90 s) using a hot plate. The filters were cooled and counted in a Microbeta scintillation counter, PerkinElmer Waltham, MA, USA).

### 4.6. Measurement of Production of cAMP

Agonist-induced changes in the cAMP level were analyzed at Gα_16__fused M_2_ and M_4_ receptors and M_2_ and M_4_ wild types. The level of cAMP was determined in radio-chromatographical separation of [^3^H]-cAMP as described previously [4]. To determine levels of cAMP, cells in suspension were pre-incubated for 1 h with 0.4 μM [^3^H]adenine, ARC (St.Louis, MO, USA), washed, and incubated for 10 min in the presence of 1 mM isobutyl methylxanthine and 10 μM forskolin. Then about 200,000 cells per 0.8 mL of sample were incubated for 1 h with tested agonists. Incubation was ended by the addition of 0.2 mL of 2.5 M HCl to the samples. Samples were applied to alumina columns (1.5 g of alumina per column, Sigma, USA), washed with 2 mL of ammonium acetate (1 M, pH = 7.0), and eluted from columns with 4 mL of ammonium acetate and measured by liquid scintillation spectrometry. Level of cAMP was expressed in dpm (decay per minute). Data are expressed as fold over basal level (after subtraction of blank value), the bottom (basal) is equal to 1.

### 4.7. Accumulation of Inositol Phosphates

The functional response of Gα_16__fused muscarinic receptors was measured as an agonist-stimulated accumulation of inositol phosphates (IPX) using radiochemical chromatography as described previously [4]. The assay was performed in cells in suspension. IPX was determined after separation on ion-exchange columns Dowex 1 × 8-200, Sigma (St.Louis, MO, USA). Harvested cells were resuspended in Krebs-HEPES buffer (KHB; 138 mM NaCl; 4mM KCl; 1.3 mM CaCl_2_; 1mM MgCl_2_; 1.2 mM NaH_2_PO_4_; 20 mM HEPES; 10 mM glucose; pH adjusted to 7.4) and centrifuged 250 g for 3 min. Cells were resuspended in KHB supplemented with 500 nM [^3^H]myo-inositol, ARC (St.Lous, MO) and incubated at 37 °C for 1 h. Then they were washed once with an excess of KHB, resuspended in KHB containing 10 mM LiCl, and incubated for 1 h at 37 °C in the presence of indicated concentrations of agonists. The total reaction volume was 800 µL. Incubation was terminated by the addition of 0.5 mL of stopping solution (chloroform: methanol: 35% HCl; 2: 1: 0.1) and placed in 4 °C for 1 h. An aliquot (0.6 mL) of the upper (aqueous) phase was taken and loaded onto ion-exchange columns. Columns were washed with 10 mL of deionized water and 20 mL of 60 mM ammonium formate/5 mM sodium borate solution. IPX were collectively eluted from columns by 4 mL of 1 M ammonium formate-0.1 M/formic acid buffer. Level of IPx is expressed in dpm (decay per minute). Data are expressed as fold over basal level (after subtraction of blank value), the bottom (basal) is equal to 1.

### 4.8. Used Agonists

Muscarinic agonists arecoline, carbachol furmethide, iperoxo, McN-A343, N-desmethylclozapine, oxotremorine, pilocarpine (Sigma, St.Louis, MO, USA), xanomeline (Tocris Bioscience, Bristol, UK), JR-6, and JR-7 (synthesized at Barry University, Miami Shores, FL, USA [19]) were used in this study. Structures of all used agonists are in the Appendix A.

### 4.9. Data and Analysis

Experiments were independent, using different seedings of CHO cells. Binding experiments were carried out in three experiments with samples in quadruplicates and functional assays were carried out at least in three experiments with samples in triplicate. Experimenters were blind to tested agonists.

After subtraction of non-specific binding (binding experiments) or background/blank values (functional experiments) data were normalized to control values determined in each experiment. IC_50_ and EC_50_ values and parameters derived from them (Ki and K_A_) were treated as logarithms. All data were included in the analysis, no outliers were excluded. In statistical analysis value of *p* < 0.05 was taken as significant for all data. In multiple comparison tests ANOVA with *p* < 0.05 was followed by Tukey HSD post-test (*p* < 0.05). Data were processed in Microsoft office, analyzed, and plotted using the program Grace. The statistic was calculated using R (www.r-project.org, accessed on 13 September 2021).

#### 4.9.1. [^3^H]NMS Saturation Binding

The equilibrium dissociation constant (K_D_) and maximum binding capacity (B_MAX_) were determined in the saturation experiments. Non-specific binding in the presence of 10 µM atropine was subtracted to determine specific binding. Free concentration of [^3^H]NMS was calculated by subtraction of values of specific binding from the final concentration of [^3^H]NMS calculated from measurements of added radioactivity. Equation (1) was fitted to the data.
(1)$$y=\frac{B_{\mathrm{MAX}}*x}{K_{D}+x}$$
where y is specific binding at free concentration x. K_D_ values are expressed as negative logarithms and B_MAX_ values as pmol of binding sites per mg of membrane protein.

#### 4.9.2. Competition Binding

The binding of tested agonists was determined in competition experiments with 1 nM [^3^H]NMS fitting of Equation (2) for one-site competition or Equation (3) for two-site competition
(2)$$y=100-\frac{100*x}{{x+\mathrm{IC}}_{50}}$$
(3)$$y=100-\frac{(100-\mathrm{flow})*x}{{x+\mathrm{IC}}_{50\mathrm{high}}}-\frac{\mathrm{flow}*x}{{x+\mathrm{IC}}_{50\mathrm{low}}}$$
where y is specific radioligand biding at concentration x of competitor expressed as a percent of binding in the absence of a competitor, IC_50_ is concentration causing 50% inhibition of radioligand binding, flow is the fraction of low-affinity binding sites expressed in percents.

Inhibition constants K_I_ for analyzed agonists were calculated as
(4)$$K_{I}=\frac{{\mathrm{IC}}_{50}}{1+\frac{\left[D\right]}{K_{D}}}$$
where IC_50_ is concentration causing 50 % inhibition of [^3^H]NMS binding calculated according to Equation (2) or (3) from competition binding data, [D] is the concentration of [^3^H]NMS used, and K_D_ is its equilibrium dissociation constant calculated according to Equation (1) from saturation binding data. Inhibition constants K_I_ are expressed as negative logarithms.

#### 4.9.3. Functional Response

The potency of analyzed agonists (EC_50_) to induce maximal response (E′_MAX_) were obtained by fitting Equation (5) to the data from measurement of the accumulation of inositol phosphates,
(5)$$y=1+\frac{\left({E^{\prime }}_{\mathrm{MAX}}-1\right){*x}^{\mathrm{nH}}}{{\mathrm{EC}}_{50}^{\mathrm{nH}}{+x}^{\mathrm{nH}}}$$
where y is a functional response at a concentration of tested compound x, E′_MAX_ is the apparent maximal response to the tested compound, EC_50_ is concentration causing half-maximal effect and ^nH^ is slope factor (Hill coefficient). EC_50_ values are expressed as negative logarithms and E_MAX_ values as folds over basal.

#### 4.9.4. Operational Model of Functional Agonism

The operational efficacy coefficient τ [77] was determined by fitting Equation (6) to data from the functional assay.
(6)$$y=\frac{E_{\mathrm{MAX}}*\tau *x}{K_{A}+\left(\tau +1\right)*x}$$
where y is a functional response at a concentration of tested compound x, E_MAX_ is the maximal response of the system, K_A_ is the equilibrium dissociation constant. Equation (6) was fitted to data from functional experiments. Equation (6) was fitted to data by the two-step procedure described earlier [31]. In the first step, system E_MAX_ was determined using carbachol, oxotremorine, and pilocarpine as internal standards by global fit to all data for a given receptor subtype and signaling pathway. In the second step, Equation (6) with E_MAX_ fixed to the value determined in the first step was fitted to individual experimental data sets.

#### 4.9.5. Relative Intrinsic Activity

For comparison of effects of agonists at different receptors fused with alpha Gα_16_ to IPX signaling pathways, relative intrinsic activity (R_Ai_) was calculated according to Griffin et al. [70].
(7)$${\mathrm{RA}}_{i}=\frac{\tau_{\mathrm{carbachol}}{*K}_{\mathrm{Aa}}}{\tau_{a}{*K}_{\mathrm{Acarbachol}}}$$
where τa and K_Aa_ are half-effective concentration and apparent maximal response to the tested compound, respectively. As Hill coefficients were equal to one, R_Ai_ values were calculated according to Equation (8).
(8)$${\mathrm{RA}}_{i}=\frac{{E^{\prime }}_{\mathrm{MAXcarbachol}}{*\mathrm{EC}}_{50a}}{{E^{\prime }}_{\mathrm{MAXa}}{*\mathrm{EC}}_{50\mathrm{carbachol}}}$$
where EC_50a_ and E′_MAXa_ are half-effective concentration and apparent maximal response to the tested compound, respectively.

#### 4.9.6. Signaling Bias

For receptors activating two or more signaling pathways, a ligand that has greater R_Ai_ value for one pathway than for other(s) is biased to that pathway. Analogically, for a single signaling pathway and two or more receptors, a ligand that has greater R_A_i at one receptor than at other(s) is biased to a given pathway at that receptor.

Analysis of signaling bias via bias factor 10^∆∆log(τ/KA)^ introduced by Kenakin et al., 2012 [35] is summarized in Appendix A.

## 5. Conclusions

The analysis of agonist bias at individual G-protein mediated pathways including non-preferential ones, plays a relevant role in the agonist screening and the development of drugs with reduced side effects that temper their clinical use. Our data showed that fusion proteins of muscarinic receptors and Gα subunits can serve as a suitable approach to analyze agonist bias and to serve as a convenient screening tool. Fusion proteins provide 1:1 receptor Gα stoichiometry, which makes quantification of agonist bias easier. We demonstrate that fusion of muscarinic receptors with Gα_16_ limits access of other competitive Gα subunits to the receptor. That, in turn, makes it easier to quantify signaling via the non-canonical Gα_16_. We demonstrated agonist-specific activation of G_16_ mediated pathway among individual subtypes of muscarinic receptors. We have confirmed functional selectivity of novel muscarinic agonists JR6 and JR7 for G_i/o_ signaling pathway [19]. Furthermore, our data revealed signaling bias of oxotremorine towards non-canonical G_16_ at M_2_ and impairment of iperoxo mediated signaling through G_16_, regarding G_i/o_ and G_q/11_ for M_2_ and G_q/11_ for M_5_ G-proteins.

## Figures and Tables

**Figure 1 ijms-22-10089-f001:**
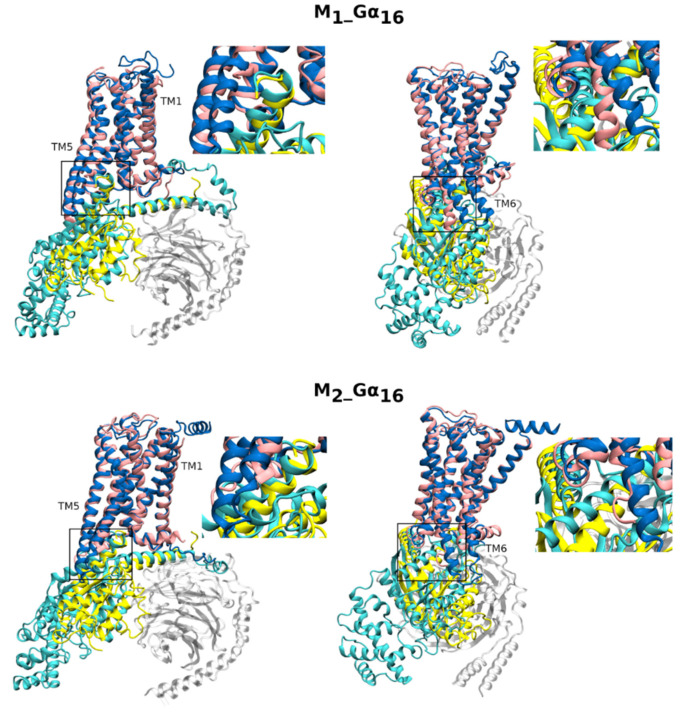
Comparison of homology models of fusion proteins with cryo-EM structures of receptor-G-protein complexes. Comparison of homology models of M_1__Gα_16_ (upper, blue_cyan) and M_2__Gα_16_ (lower, blue_cyan) fusion proteins with cryo-EM structures of M_1_ receptor in an active conformation induced by iperoxo (upper, pink) in complex with Gα_11_ (upper, yellow) (6OIJ) and M_2_ receptor in an active conformation induced by iperoxo (lower, pink) in complex with Gα_oA_ (lower, yellow) (6OIK) as viewed TM4 and TM5 (left) or TM6 and TM7 (right) in front. Complexes of βγ-subunits of G-proteins from cryo-EM structures are shown in grey. Structures were aligned on the receptor molecule. Details of insertion of C-terminus of α-subunit into G-protein binding site of the receptor are enlarged in the insets.

**Figure 2 ijms-22-10089-f002:**
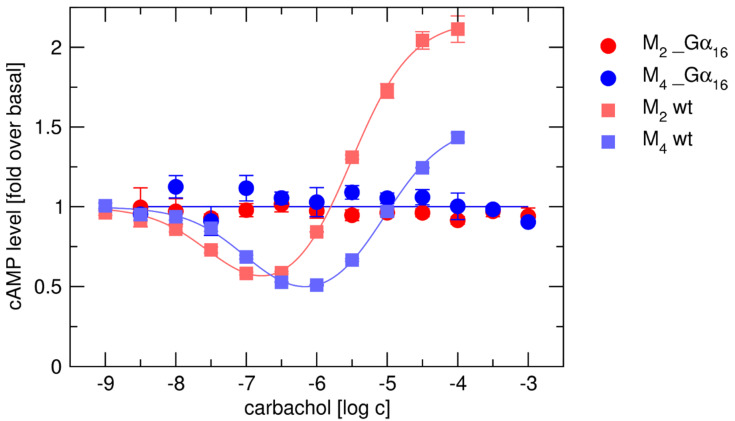
Carbachol-stimulated changes in the cAMP level. Changes in the forskolin-stimulated level of cAMP were measured at CHO cells expressing wt (squares) or Gα_16__fused (circles) M_2_ (red) and M_4_ (blue) receptors after stimulation by increasing concentration of carbachol. Data are expressed as fold over the basal level of cAMP (in absence of carbachol). Basal level of cAMP was determined in presence of 10 µM forskolin and is equal to 1. Data are means ± SD from three independent experiments performed in triplicate.

**Figure 3 ijms-22-10089-f003:**
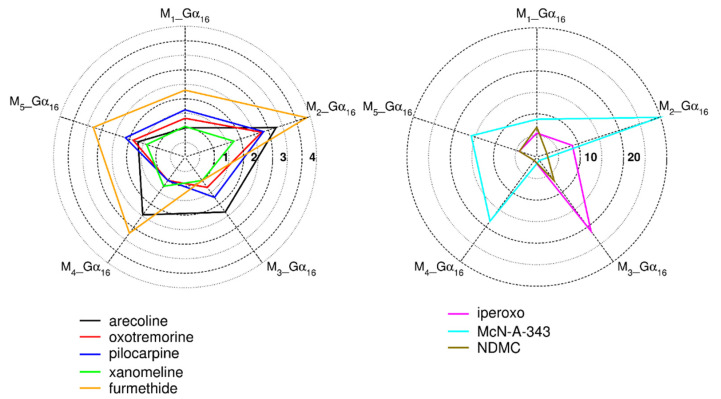
Polar plot of relative intrinsic activity RA_i_. Intrinsic activities of individual agonists relative to reference agonist carbachol (RAi) calculated according to Equation (8) from the measurement of the accumulation of inositol phosphates are plotted. Values are expressed as ratios of RAi to RAi at receptor with the lowest activity for given agonist (Arecoline M1; Furmethide, McN-A-343, Xanomeline M_3_; NDMC, Oxotremorine, Iperoxo, Pilocarpine M_4_).

**Figure 4 ijms-22-10089-f004:**
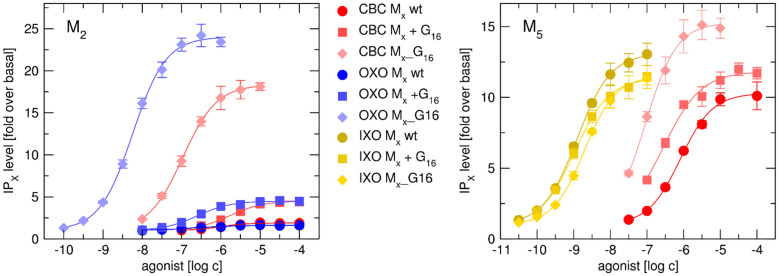
Comparison of functional response of M_2_ and M_5_ receptor variants to agonists. Accumulation of inositol phosphates (IPx) induced by increasing concentration of agonists carbachol (CBC-red), oxotremorine(OXO-blue), or iperoxo (IXO-yellow) in CHO cells expressing wt (circles), Gα_16_ subunit co-transfected (diamonds) or Gα_16__fused (squares) M_2_ (left graph) and M_5_ (right graph) receptors. Data are expressed as folds over the basal level (in absence of agonist) and bottom is equal to 1. Data are means ± SD from three independent experiments performed in triplicate.

**Table 1 ijms-22-10089-t001:** Affinities of muscarinic agonists to Gα_16__fused receptors Affinities of muscarinic agonists are expressed as negative logarithms of inhibition constants (K_I_) of [^3^H]NMS binding to individual subtypes of muscarinic receptors fused with Gα_16_-subunit. They were calculated according to Equation (4) from IC_50_ values obtained by fitting Equation (2) or (3) to data from competition experiments with [^3^H]NMS. Values are means ± SD from three independent experiments performed in quadruplicates.

	M_1__Gα_16_	M_2__Gα_16_	M_3__Gα_16_	M_4__Gα_16_	M_5__Gα_16_
Arecoline	5.19 ± 0.06	4.68 ± 0.03	5.17 ± 0.08	4.68 ± 0.01	5.16 ± 0.04
Carbachol	4.87 ± 0.01	4.62 ± 0.01	4.77 ± 0.02	4.61 ± 0.02	4.72 ± 0.01
Furmethide	5.79 ± 0.01	4.69 ± 0.04	5.27 ± 0.03	4.71 ± 0.01	5.25 ± 0.02
Iperoxo (high)	8.35 ± 0.12	n.d.	n.d.	n.d.	n.d.
Iperoxo (low)	6.20 ± 0.08	5.83 ± 0.03	6.06 ± 0.04	5.96 ± 0.03	6.99 ± 0.02
McN-A-343	4.24 ± 0.04	6.54 ± 0.04	5.14 ± 0.02	6.41 ± 0.02	5.34 ± 0.06
NDMC	7.06 ± 0.01	6.51 ± 0.04	6.75 ± 0.02	6.40 ± 0.01	6.77 ± 0.03
Oxotremorine	6.61 ± 0.01	5.70 ± 0.04	6.24 ± 0.03	5.86 ± 0.02	6.16 ± 0.03
Pilocarpine	5.26 ± 0.02	4.52 ± 0.01	4.92 ± 0.02	4.54 ± 0.03	4.88 ± 0.04
Xanomeline	7.29 ± 0.01	6.82 ± 0.02	7.19 ± 0.04	7.04 ± 0.03	7.06 ± 0.02
JR-6 (high)	n.d.	n.d.	n.d.	6.73 ± 0.28	n.d.
JR-6 (low)	4.97 ± 0.07	5.74 ± 0.10	5.07 ± 0.04	5.29 ± 0.21	5.44 ± 0.05
JR-7	4.34 ± 0.05	5.17 ± 0.07	4.22 ± 0.06	4.82 ± 0.03	4.46 ± 0.04

n.d., not determined.

**Table 2 ijms-22-10089-t002:** Parameters of functional response of Gα_16__fused receptors. Operational efficacy τ, agonist equilibrium dissociation constant K_A,_ and agonist relative intrinsic activity R_Ai_ were calculated according to Equations (6)–(8), respectively, from parameters of functional response EC_50_ and E´_MAX_ (Appendix A) obtained by fitting Equation (5) to data from measurement of the accumulation of inositol phosphates. Values of system E_MAX_ are (27.1 ± 0.5 for M_1__Gα_16_; 30.7 ± 0.603 for M_2__Gα_16_; 27.1 ± 0.6 for M_3__Gα_16_; 27.1 ± 1.2 for M_4__Gα_16_; and 28.9 ± 0.4 for M_5__Gα_16_). K_A_ is expressed as negative logarithms. Values are means ± SD from three independent experiments performed in triplicate.

		Arecoline			Carbachol			Furmethide			Iperoxo			McN-A343			NDMC			Oxotremorine			Pilocarpine			Xanomeline			JR6	JR7
M_1__G_16_	τ	0.594	±	0.057	0.887	±	0.017	0.655	±	0.005	1.577	±	0.055	0.472	±	0.02	0.5	±	0.009	0.869	±	0.027	0.668	±	0.04	0.795	±	0.007	0	0
	pK_A_	6.68	±	0.01	6.8	±	0.06	6.3	±	0.03	8.24	±	0.03	6.53	±	0.1	7.43	±	0.05	7.82	±	0.05	6.29	±	0.02	8.25	±	0.02	n.c.	n.c.
	RA_i_	0.45	±	0.025	1	±	0.01	0.234	±	0.001	45.9	±	0.9	0.248	±	0.006	2.4	±	0.02 *	10.3	±	0.2	0.234	±	0.008	25	±	0.1	0	0
M_2__G_16_	τ	1.566	±	0.038	1.41	±	0.028	1.669	±	0.034	9.61	±	0.227	1.033	±	0.021	0.951	±	0.016	2.799	±	0.081	0.946	±	0.02	1.669	±	0.034	0	0
	pK_A_	6.73	±	0.06	6.6	±	0.06	6.18	±	0.03	7.62	±	0.08	6.68	±	0.04	6.7	±	0.02	7.64	±	0.08	6.39	±	0.02	8.15	±	0.03	n.c.	n.c.
	RAi	1.47	±	0.02 *	1	±	0.01	0.446	±	0.005 *	71.3	±	1	0.875	±	0.010 *	0.846	±	0.008	21.4	±	0.4 *	0.407	±	0.005 *	41.9	±	0.5 *	0	0
M_3__G_16_	τ	0.541	±	0.004	0.918	±	0.02	0.656	±	0.009	2.926	±	0.876	0.307	±	0.046	0.497	±	0.003	0.834	±	0.027	0.697	±	0.039	0.777	±	0.022	0	0
	pK_A_	7.1	±	0.01	6.8	±	0.08	5.95	±	0.31	8.56	±	0.1	5.81	±	0.13	7.43	±	0.06	7.84	±	0.05	6.31	±	0.01	8.25	±	0.03	n.c.	n.c.
	RAi	1.05	±	0	1	±	0.01	0.102	±	0.001	176	±	30 *	0.028	±	0.002	2.35	±	0.01	10	±	0.2	0.247	±	0.008	23.8	±	0.4	0	0
M_4__G_16_	τ	0.804	±	0.071	0.897	±	0.04	0.865	±	0.084	1.273	±	0.055	0.704	±	0.07	0.623	±	0.05	0.994	±	0.049	0.557	±	0.054	0.866	±	0.084	0	0
	pK_A_	7.19	±	0.04	7.1	±	0.01	6.64	±	0.01	7.87	±	0.03	6.93	±	0.02	6.8	±	0.02	7.95	±	0.01	6.47	±	0.01	8.59	±	0.01	n.c.	n.c.
	RAi	1.11	±	0.06	1	±	0.03	0.33	±	0.018	8.26	±	0.21	0.523	±	0.03	0.346	±	0.016	7.76	±	0.22	0.144	±	0.008	29.6	±	1.7	0	0
M_5__G_16_	τ	0.51	±	0.005	1.126	±	0.015	1.072	±	0.01	0.825	±	0.031	0.358	±	0.009	0.709	±	0.009	1.555	±	0.016	0.8	±	0.023	1.172	±	0.015	0	0
	pK_A_	7.02	±	0.14	6.7	±	0.02	6.25	±	0.03	8.46	±	0.14	6.96	±	0.11	7.07	±	0.03	7.72	±	0.03	6.34	±	0.01	8.2	±	0.01	n.c.	n.c.
	RAi	0.76	±	0.005	1	±	0.01	0.336	±	0.002	35	±	1	0.452	±	0.006	1.47	±	0.01	14.5	±	0.1	0.309	±	0.005	33	±	0.2	0	0

n.c., not calculated; *, greater than at other subtypes (*p* < 0.05, according to ANOVA and Tukey-HSD post-test).

**Table 3 ijms-22-10089-t003:** Comparison of parameters of functional response of variants of M_2_ and M_5_ receptor. Parameters of agonist-induced functional response EC_50_ and E´_MAX_ were obtained by fitting Equation (6) to data from measurement of the accumulation of inositol phosphates. Operational efficacy τ, agonist equilibrium dissociation constant K_A_ and agonist relative intrinsic activity RA_i_ were calculated according to Equations (6)–(8), respectively. EC_50_ and K_A_ are expressed as negative logarithms. Values of system E_MAX_ are (30.7 ± 0.603 for M_2__Gα_16_; 29 ± 3 for M_5__Gα_16_; 5.8 ± 0.4 for M_2_ + Gα_16_; 21 ± for M_5_ + Gα_16_; 5.5 ± 0.4 for wt M_2_; 22 ± 2 for wt M_5_). Values are means ± SD from 3 independent experiments performed in triplicate.

		pEC50			E′MAX			τ			pKA			RAi		
M_2_+Gα_16_	carbachol	5.59	±	0.12	4.62	±	0.37	0.88	±	0.17	5.52	±	0.17 ^†^	1	±	0.03
	oxotremorine	6.53	±	0.13	4.46	±	0.33	0.85	±	0.15	6.26	±	0.10 ^†^	5.41	±	0.05 ^†^
M_2__Gα_16_	carbachol	6.99	±	0.06	18	±	0.4	1.41	±	0.03	6.6	±	0.06	1	±	0.01
	oxotremorine	8.22	±	0.08	22.4	±	0.6 *	2.8	±	0.08 *	7.64	±	0.08	21.4	±	0.4
M_2_ wt	carbachol	6.01	±	0.04	1.91	±	0.07	0.2	±	0.1	5.9	±	0.1	1	±	0.01
	oxotremorine	6.68	±	0.05	1.60	±	0.05	0.2	±	0.1	6.6	±	0.1	3.08	±	0.28
M_5_+Gα_16_	carbachol	6.61	±	0.08	11.7	±	0.4	0.814	±	0.03	6.35	±	0.08 ^†^	1	±	0.02
	iperoxo	8.95	±	0.14	11.4	±	0.8	0.785	±	0.057	8.7	±	0.14 ^†^	213	±	9 ^†^
M_5__Gα_16_	carbachol	7.03	±	0.02	15.3	±	0.2	1.126	±	0.015	6.7	±	0.02	1	±	0.01
	iperoxo	8.72	±	0.14	11.3	±	0.4 *	0.825	±	0.031 *	8.46	±	0.14	35	±	1
M_5_ wt	carbachol	6.09	±	0.16	10.1	±	1.1	0.68	±	0.077	5.86	±	0.16	1	±	0.06
	iperoxo	8.93	±	0.16	13	±	1.1 *	1.08	±	0.09 *	8.61	±	0.16	912	±	45 ^†^

*, different from carbachol (*p* < 0.05), ^†^ different from fusion protein (*p* < 0.05), according to ANOVA and Tukey HSD post-test.

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
