# Peer review of "Fusion with Promiscuous Gα16 Subunit Reveals Signaling Bias at Muscarinic Receptors"

_ijms, 2021, doi:10.3390/ijms221810089_

Round 1
Reviewer 1 Report
The work is well designed and described. Scientific results are of wide interest for researchers interested in detailed studies of biased signaling and it surely goes beyond just muscarinic receptors. This reviewer has no further critical comments.
Author Response
We thank reviewer 1 for the review of our manuscript. The native speaker Prof John Boulos approved the manuscript.
Reviewer 2 Report
In this manuscript by Randakova, the authors investigate non-canonical G16 coupling to the muscarinic receptors. The authors employ a fusion strategy incorporating fused G16 to the C-terminus of the muscarinic subtypes and conclude that oxotemorine is biased towards the G16 pathway.
However I have several issues with the manuscripts premise, presentation of the data, and approach detailed below.
- Why did the authors choose to study G16 subtype? It seems there isn’t a clear link between muscarinic receptor activation and the G16-mediated pathway. The authors provide some context for this pathway in the discussion on lines 479-481 as being involved in cell proliferation or differentiation, but I fail to see why the authors did not follow up with any measures of cell proliferation with these constructs to validate further.
- The approach of fusing effectors to the C-terminus is not a novel approach. But I doubt the claim that these receptor constructs do not couple to endogenous G proteins in the CHO cells.
For example, why did the authors not perform cAMP or IPx experiments with Pertussis toxin (PTX) to uncouple Gi/o proteins from the M2 and M4 receptors to show G16 coupling specificity? In other words, the authors have not conclusively ruled out the possibility that Gi/o proteins are still coupling to these receptor constructs, and therefore influencing the signaling data.
Furthermore, the same rationale could be said about the M1/M3/M5-G16 fused constructs in coupling to Gq. Several commercially available reagents are available to selectively uncouple Gq/11 proteins (PMID 15339913, 26658454, and 25036778). Using selective Gq/11 inhibition would likely lead to interpretations that the responses are indeed a result from the G16-fused constructs.
- The presentation of the data is lacking. For example, where is the dose-response data for the M1/M3/M5 fused constructs? Table 1 formatting is also an issue, and is difficult to interpret.
Very important, presentation of the data comparing the unfused and fused in the data tables is needed in the main part of the manuscript. Some of the binding data is presented in the supplement, but where is the functional data for the unfused comparing to the fused constructs? Without presentation of the dose-response data for the fused and unfused constructs for various agonist, it is very difficult if the parameter estimates of functional activity make sense.
Again it is very difficult to make any conclusions about the coupling efficiency of the G16 fused constructs if no meaningful controls were taken to uncouple endogenous G proteins in the CHO cell system.
Also by the way, the data presented in Fig 4 shows there is no clear bottom to the carbachol M5-G16 construct or G16 transfected condition. How can the authors fit these nonlinear regression data without a clear bottom?
- Exploration of the high affinities comparing the fused G16 to the unfused muscarinic receptors is also lacking. High affinity estimates would further support the authors’ claims of bias by demonstrating an increased preference of oxtremorine for the G16 fused construct over the unfused construct.
Furthermore it is worrisome that none of the raw binding data curves are presented in the manuscript. The shallow shape of the high affinity binding curve is key for the conclusion that oxotremorine is shifting the G16 fused construct more into the active state. Please see this example for this fusion approach and analysis to detect high affinity shifts with specific fused effectors (PMID 24668815).
- Did the authors do any controls to ensure that basal activity of the G16 fused constructs was similar or equal to the unfused constructs for IPx generation? The authors only show fold-over-basal but what about absolute basal activity and what about absolute Emax for this response. Indeed, it seems the unfused and fused constructs expression levels by Bmax estimation in Table S1 differ substantially. Comparing Bmax for M2-G16 versus M2 unfused, the unfused expresses approximately 17-fold higher! I would expect the functional activity and shifts in EC50 to differ substantially between these two conditions, and this can impair the conclusions of any bias.
- Finally, I would also strongly recommend the authors to use the fusion strategy for the “canonical” G proteins for the individual receptors (i.e. M5-Gq) as an important control. Extending the fusion approach to these canonical G proteins will make for similar comparisons between constructs, rather than relying on endogenous CHO proteins versus the G16-fused construct for comparison. For example, why did the authors not “pilot” test this fusion strategy with M2-Gi1 or other Gi/o subtype to see indeed coupling can occur? At the very least this serves as important “proof of principle” to show coupling can be specific and controlled using this approach. Then why not use that canonical fusion construct for comparison to the G16-fused construct to determine bias. Again relying on endogenous CHO G protein couplings compared to a fused construct makes the biased agonist comparison flawed.
Author Response
We thank the reviewer for constructive criticism and suggestions. We substantially rewrote the manuscript, mainly the Results and Discussion to improve clarity as described in detail in the attached file.

Reviewer 3 Report
In this manuscript Randáková and colleagues describe the construction and pharmacologic testing of muscarinic receptors 1-5 that have been fused to the promiscuous Ga16 protein. Through this approach they studied agonist bias for this non-preferential G protein without interference from other more favored G proteins. To this end they tested binding to and functional response of 11 different agonists on the GPCR/G protein-fusion proteins. Selected agonists were then used to directly compare wt receptor alone, with co-transfected G protein and the fusion receptor. Further, agonist bias for all 5 fusion receptors was explored for all activating agonists.
In general, this is an interesting approach, which can be used to study additional signaling through G proteins that would normally be overrun by the signaling from the preferred G proteins. As such new signaling bias could be established using this knowledge. However, I am not sure how relevant this would actually be in vivo, as here the cell incorporates all inputs. The authors mention that tissue-specific expression of G proteins could lead the way, yet I did not see where the used G16 protein would play role (maybe I missed this). I am only mentioning this because the authors point directly at a disease/treatment relevance of their finding, while I believe that also simply the theoretical addition of knowledge would be interesting enough to warrant publication.
Ideally the authors discuss the physiological importance of non-preferential coupling and specifically, how much input can be expected from this when dominant G proteins are present.
I am reluctant to judge the binding assays, homology modeling as well as the different equations that have been used, as I have not been performing these types of experiments/calculations myself. Regarding the cAMP and IPX signaling assays, I would appreciate it if the authors would indicate at the respective methods sections how long the agonists were incubated.
Apart from that I feel that the data presentation is a bit confusing and inconsistent, even though probably all the data is there (as one can deduct from the tables).
In detail:
- 2: What are the basal cAMP levels? Is basal after stimulation with forskolin? Please indicate (e.g. in the figure legend) what are the nM cAMP levels that represent 1. Please provide data on mock control (supplements would be fine). For consistency’s sake it would be good to add data on M2 and M4, each plus co-transfected G16 protein.
- 4 and 5 appear to be redundant to the data shown in Tab. 3 and it is not clear why this information comes after the bias analysis in Fig. 3. I appreciate showing actual concentration response curves in addition to the Table summary but this should go into the same paragraph. I personally think that Figs. 2, 4 and 5 should be displayed in the same figure, followed by Table 3 and be reported in the same paragraph.
- The results section would in general benefit from more subheadings in the ‘Functional response of fusion proteins to agonists’ section.
- 5: Similar to Fig. 2 I am missing the comparison to M2 plus G16 for both agonists.
- 1: What is the purpose of all the numbers (why does it begin with 7.?)
- 3: I am missing data on M2 wt and any mentioning of M3 and M4. This would be essential as this table is at the heart of the study.
Minor points:
- The reference #6 (Grundmann et al 2018) does not fit to the sentence as this paper specifically excludes GPCR-signaling through arrestins.
- Please check grammar. Ideally, have a native speaker look at the manuscript.
- Please, shorten the discussion. There is no need to list all of the results once more and several points are being mentioned multiple times.
Author Response

(The authors gave the same response as above.)
